# Comparative Physiological and Transcriptomic Analyses of Two Contrasting Pepper Genotypes under Salt Stress Reveal Complex Salt Tolerance Mechanisms in Seedlings

**DOI:** 10.3390/ijms23179701

**Published:** 2022-08-26

**Authors:** Tao Zhang, Kaile Sun, Xiaoke Chang, Zhaopeng Ouyang, Geng Meng, Yanan Han, Shunshan Shen, Qiuju Yao, Fengzhi Piao, Yong Wang

**Affiliations:** 1College of Horticulture, Henan Agricultural University, Zhengzhou 450002, China; 2Henan Academy of Agricultural Sciences, Zhengzhou 450002, China; 3College of Plant Protection, Henan Agricultural University, Zhengzhou 450002, China

**Keywords:** pepper, salt stress, physiological analysis, RNA-seq

## Abstract

As a glycophyte plant, pepper (*Capsicum annuum* L.) is widely cultivated worldwide, but its growth is susceptible to salinity damage, especially at the seedling stage. Here, we conducted a study to determine the physiological and transcriptional differences between two genotype seedlings (P300 and 323F3) with contrasting tolerance under salt stress. The P300 seedlings were more salt-tolerant and had higher K^+^ contents, higher antioxidase activities, higher compatible solutes, and lower Na^+^ contents in both their roots and their leaves than the 323F3 seedlings. During RNA-seq analysis of the roots, more up-regulated genes and fewer down-regulated genes were identified between salt-treated P300 seedlings and the controls than between salt-treated 323F3 and the controls. Many ROS-scavenging genes and several SOS pathway genes were significantly induced by salt stress and exhibited higher expressions in the salt-treated roots of the P300 seedlings than those of 323F3 seedlings. Moreover, biosynthesis of the unsaturated fatty acids pathway and protein processing in the endoplasmic reticulum pathway were deeply involved in the responses of P300 to salt stress, and most of the differentially expressed genes involved in the two pathways, including the genes that encode mega-6 fatty acid desaturases and heat-shock proteins, were up-regulated. We also found differences in the hormone synthesis and signaling pathway genes in both the P300 and 323F3 varieties under salt stress. Overall, our results provide valuable insights into the physiological and molecular mechanisms that affect the salt tolerance of pepper seedlings, and present some candidate genes for improving salt tolerance in pepper.

## 1. Introduction

Affecting agricultural systems around the globe, salinity is a prevalent abiotic stress that affects more than 20% of the global cultivated land (about 300 million hectares) and limits plant growth, productivity, and distribution; it arises mainly as a result of irrigation with poor quality water and soil salinization [1,2,3]. With the world’s population constantly increasing, improving plants’ salt tolerance is essential to meeting the growing demand for food [4]. Investigating the mechanisms of crop responses to salt stress could improve our understanding of the genetic basis of salt tolerance and provide the basis for effective engineering strategies to improve salt tolerance [5], starting with the reconsideration of wild donors, known as crop wild relatives (CWRs) [6,7,8,9]. Salt-tolerant genotypes are ideal gene donors for studying the tolerance mechanisms of a particular crop and improving its salt tolerance.

Salt stress hinders the homeostasis of K^+^/Na^+^ in the plant cytosol, resulting in a decrease in the K^+^/Na^+^ ratio and damage to the selectivity of root membranes [10]. To prevent growth cessation or cell death, plants adopt many adaptive mechanisms to confront salt stress [1,11]. For example, to cope with salt stress, plants extrude excessive Na^+^ or compartmentalize it in the vacuole by various mechanisms, such as the salt overly sensitive (SOS) pathway [11]. Another adaptive mechanism is the accumulation of compatible solutes, which are important for plant osmo-regulation and osmo-tolerance under salt stress [11]. Increased endogenous compatible solutes, including proline, soluble sugar, and soluble protein are related to enhanced salt tolerance in many plants [12,13,14,15]. Abiotic stresses usually lead to protein dysfunction in plant cells, which can be effectively alleviated by heat-shock proteins (HSPs) [16,17]. Unsaturation levels of membrane fatty acids, which are mainly controlled by fatty acid desaturases (FADs), are also associated with salt tolerance in *Arabidopsis thaliana* (L.) Heynh. [18]. 

Plant responses to salt stress are also mediated by many phytohormones [19]. Ethylene (ETH) is involved in plant salt stress, but its roles in salt tolerance are complex [20,21]. The overexpression of an ethylene response factor (*ERF*) gene from soybean [*Glycine max* (L.) Merr.] in tobacco (*Nicotiana tabacum* L.) can enhance its salt tolerance [22]. ETH production in maize (*Zea mays* L.) is divided into two stages during NaCl treatment: the first stage (Phase I) is required for salt tolerance, whereas the second stage (Phase II) results in salt sensitivity [21]. Exogenous jasmonic acid (JA) can increase salt tolerance in wheat (*Triticum aestivum* L.) seedlings [23], and the JA signaling pathway is also activated by salt stress in *Arabidopsis* roots [23], indicating that JA is an important hormone in plant response to salt stress. Abscisic acid (ABA) is also reported to be accumulated in plant roots and affects lateral root development under salt stress [19]. Furthermore, growth hormones such as auxin/indole-3-acetic acid (AUX/IAA) play an important role in allowing to cope with salt stress by regulating plant growth and development [19].

Salinity usually results in the generation of excessive reactive oxygen species (ROS), leading to oxidative damage to lipids, protein, and nucleic acids in plants [11,24]. ROS are reactive forms of molecular oxygen, mainly comprising singlet oxygen (^1^O_2_), superoxide anion (O_2_^−^), hydrogen peroxide (H_2_O_2_), and hydroxyl radical (HO·) [25,26]. H_2_O_2_ in peroxisomes accounts for about 70% the total ROS pool in photosynthesizing tissues [25]. To scavenge ROS and protect cells from potential cytotoxic effects, plants have developed an effective antioxidant system, including enzymatic and non-enzymatic mechanisms, to react with the excessive ROS and keep ROS at a low level [26,27]. Superoxide dismutase (SOD), catalase (CAT), peroxidase (POD), ascorbate peroxidase (APX), and glutathione S-transferase (GST) are ubiquitous antioxidant enzymes, and glutathione (GSH) and ascorbate (AsA) are important non-enzymatic antioxidants in plants [26]. In addition, thioredoxin (Trx)/peroxiredoxin (Prx) and GSH-dependent Glutaredoxin (Grx) systems also play important roles in regulating redox homeostasis under oxidative stress [26]. Salt stress exhibits different impacts on the components of antioxidant defense systems in plants [26,27,28].

Pepper (*Capsicum annuum* L.), one of the most widely grown vegetables globally, is a glycophyte plant. Salt stress has a significantly negative effect on pepper growth, and different genotypes of pepper show different responses to salinity [29]. Although salt-response mechanisms have been thoroughly investigated by studying model plants, the most suitable salt tolerance mechanisms differ between species because of their different genetic backgrounds [1]. Thus far, the physiological responses of pepper to salt stress have been expounded in several studies [29,30,31,32]; however, these studies offer little information regarding the underlying molecular regulatory mechanisms of salt tolerance in pepper. This study compared the performance and physiological responses to salt stress of two genotypes (a salt-sensitive pepper ‘323F3’ and a salt-tolerant pepper ‘P300’). Transcriptome analysis has been widely used in identifying candidate genes related to abiotic stress responses [33,34]. Plant roots are directly exposed to abiotic stresses in the soil. Thus, to enhance our understanding of salt-tolerance mechanisms in pepper, the root transcriptome profiles in the two genotypes were further investigated both under normal conditions and under salt stress.

## 2. Results

### 2.1. Performance of 323F3 and P300 under Salt Stress

In this study, 150 mM NaCl was used to assess the differences in salt tolerance between 323F3 and P300. Our results showed that salt stress inhibited seedling growth and leaf surface expansion; this resulted in downward leaf curling in both genotypes, but it was more serious in 323F3 (Appendix A and Figure 1A). After 20 days of salt treatment, the plant height (PHE), stem diameter (SDA), fresh weight (SFW), and dry weight (SDW) of the shoots (323F3/P300) decreased by 27.26%/15.22%, 21.40%/18.43%, 59.02%/47.15% and 47.67%/39.60% (Figure 1B), respectively, meaning that the shoot growth of the two genotypes was inhibited by salt stress. Based on these results, the salt-stress-tolerance indices of PHE and SFW in P300 were significantly higher than those in 323F3, indicating that the shoot of P300 were more able to adapt salt stress. Moreover, in terms of their roots, the length (RL), surface area (RSA), diameter (RDA), volume (RVO), root vigor (RVI), fresh weight (RFW), and dry weight (RDW) of 323F3/P300 were reduced by 32.19%/8.17%, 38.25%/10.18%, 36.07%/13.18%, 75.69%/13.87%, 46.65%/21.68%, 56.54%/24.27% and 59.86%/37.91% (Figure 1C), respectively, meaning that salt stress also had the obvious effect of inhibiting root growth in the two genotypes. Noticeably, each parameter in the P300 roots exhibited a higher salt stress tolerance index than those of the 323F3 roots (Figure 1C), indicating that the roots of P300 also had a better ability to cope with salt stress. Overall, these data showed that P300 was more tolerant to salt stress than 323F3.

### 2.2. Physiological Responses to NaCl Stress in 323F3 and P300

The ion balance is highly important for plants’ salt tolerance [10]. Therefore, the concentrations of Na^+^ and K^+^ were measured in both the shoots and the roots of the two genotypes (Figure 2). The results showed that NaCl treatment significantly elevated Na^+^ concentrations (Figure 2A,E) and Na^+^/K^+^ ratios (Figure 2C,G) in the shoots and roots of the two genotypes, but the increases were more dramatic in 323F3 than in P300. Salt stress decreased the (Na^+^) shoot/(Na^+^) root ratio in both 323F3 and P300, but there were no apparent differences between the genotypes regardless of salt stress (Figure 2D), suggesting that the genotypic difference in the translocation of Na^+^ from the roots to the shoots might not be significant. Additionally, the K^+^ concentration in 323F3 distinctly decreased under salt stress. Especially in its roots, P300 exhibited a relatively stable K^+^ concentration during salt treatment, which resulted in a higher K^+^ content in P300 than in 323F3 under salt stress (Figure 2B,F). Taken together, these results indicated that P300 exhibited a better ability than 323F3 to maintain its ion balance and reduce Na^+^ toxicity under salt stress, which may help to improve its salt tolerance.

Antioxidant enzymes (e.g., SOD, CAT, and POD) are beneficial for alleviating abiotic-stress-triggered ROS bursts to enhance plant tolerance [26]. Our results indicated that salt stress resulted in an obvious elevation in the activities of different antioxidant enzymes in pepper, but the performance was different in 323F3 and P300 (Figure 3). For 323F3, the POD activity in leaves/roots increased by 328.73%/61.35% (Figure 3B,E), whereas CAT and SOD were significantly induced only in the leaves, by about 21.85% and 59.92% (Figure 3A,C), respectively, under salt stress. For P300, the activities of CAT, POD, and SOD were significantly enhanced by salt stress in both the leaves and the roots, and the increment rates in the leaves/roots were 27.42%/43.92%, 336.41%/12.98%, and 106.46%/26.57%, respectively. Notably, the activities of the three antioxidant enzymes in P300 leaves were significantly higher than those of the 323F3 leaves, and SOD and CAT also exhibited higher activities in P300 roots than in 323F3 roots.

The accumulation of compatible organic solutes under control conditions and under salt treatment was also detected. The results indicated that the contents of proline, soluble sugar, and soluble protein in the two genotypes were significantly altered by salt stress; this effect was especially pronounced in P300, and was different in leaves and roots. Under control conditions, there were almost no significant differences in the contents of proline, soluble sugar, and soluble protein between the leaves of 323F3 and P300 (Figure 4A–C). When salt stress was applied, the contents of proline and soluble sugar increased and were clearly higher in P300 leaves than in 323F3 leaves, but the levels of soluble protein were affected very little. In the roots, the contents of proline, soluble sugar, and soluble protein were similar, lower and higher in P300 compared with 323F3, respectively, under control conditions (Figure 4B–D). Their contents were significantly increased by salt stress in P300 roots, but only proline was observed to increase in 323F3 roots under salt stress. In addition, the contents of the three solutes were obviously higher in P300 roots than in 323F3 roots under salt stress. These findings suggested that P300 accumulated more compatible organic solutes in its leaves and roots under salt stress than 323F3.

To assess the effects of salt stress on membrane lipid peroxidation and membrane permeability, the malondialdehyde (MDA) content and the relative electric conductivity (REC) of leaves were determined (Figure 5). Our results show that, without NaCl stress, the MDA and REC levels in 323F3 leaves were no different from those of the P300 leaves. Salt stress led to significant increases in the levels of MDA and REC in both genotypes, but the increases were more pronounced in 323F3 (increased by 84.21% for MDA and 35.00% for REC) than in P300 (increased by 33.81% for MDA and 17.49% for REC), suggesting that, under salt stress, the cell membrane was more integrated in P300 than in 323F3.

### 2.3. RNA-seq Analysis and Identification of DEGs

To better understand the different mechanisms of 323F3 and P300 in response to salt stress, RNA-seq analysis of their roots was performed under normal and salt-stress conditions. In total, 7910 genes were identified as differentially expressed genes (DEGs) in at least one comparison; they were divided into two major clusters in 323F3 and P300 (Appendix A). A heatmap shows that the overall gene expression levels were increased by salt stress, especially in P300 (Appendix A). Moreover, subclusters between the control and salt treatments and subclusters between the two genotypes within the three biological replicates of each sample were also found (Appendix A), meaning that the expression patterns of the replicates were highly correlated. Principal component analysis indicated that the first and second variance were 70% and 20%, respectively. Additionally, the three biological replicates of each sample clustered together, which further confirmed their high correlation (Appendix A). In addition, we randomly selected 10 genes to validate the RNA-Seq data by qRT-PCR. As shown in Appendix A, the expression patterns obtained by qRT-PCR were in good agreement with the changes in their transcript abundance as identified by RNA-seq (R^2^ = 0.9127). Thus, the transcriptome data were reproducible, reliable, and suitable for further study.

Before salt stress, we found 1417 up-regulated and 2541 down-regulated genes in P300 compared with 323F3 (323F3C vs. P300C) (Figure 6A). After 48 h of salt exposure, 2048 up-regulated and 1972 down-regulated genes were identified in the 323F3S vs. P300S comparison (323F3S as the control) (Figure 6A). Moreover, in 323F3, 556 and 1257 genes were up- and down-regulated by salt stress, respectively (323F3C vs. 323F3S, 323F3C as the control); in P300, 1597 and 1067 genes were up- and down-regulated by salt stress, respectively (P300C vs. P300S, P300C as the control) (Figure 6A). These results showed that a relatively larger number of genes were activated by salt stress in P300 compared with 323F3.

DEGs in the four comparisons were evaluated to study their function in response to the salt stress. Figure 6B shows that the responsive genes can be divided into two main types: genotype-specific and common salt-stress responsive genes. There were 532 (P300C vs. P300S) and 416 (323F3C vs. 323F3S) genotype-specific responsive genes and 599 common salt-stress responsive genes (323F3C vs. 323F3S and P300C vs. P300S). Taking into account the background differences between the two genotypes, we mainly focused on the 323F3C vs. 323F3S and P300C vs. P300S comparisons in the subsequent transcriptome analysis.

### 2.4. Gene Ontology Enrichment Analysis of DEGs

Gene ontology (GO) enrichment analysis was performed to predict the roles of the 323F3C vs. 323F3S and P300C vs. P300S comparisons. The results showed significant differences in many of the over–represented GO terms between the two comparisons (Figure 7). For the biological processes, terms related to stress response and tolerance, such as defense response (GO:0006952), oxidation–reduction process (GO:0055114), hydrogen peroxide catabolic process (GO:0042744), response to stress (GO:0006950), cell wall macromolecule metabolic process (GO:0044036), and reactive oxygen species metabolic process (GO:0072593), were found to be highly enriched in the P300C vs. P300S comparison, but not in the 323F3C vs. 323F3S comparison. Most of the DEGs involved in defense response, oxidation–reduction process, response to stress, and cell wall macromolecule metabolic process were up-regulated by salt stress in P300. In terms of the cellular components, many terms related to the component of membrane and plasma membrane, such as the intrinsic component of membrane (GO:0031224), integral component of membrane (GO:0016021), integral component of plasma membrane (GO:0005887) and intrinsic component of plasma membrane (GO:0031226), were enriched in both of the two comparisons. Most of the DEGs involved in intrinsic and integral components of the membrane were up-regulated in the P300C vs. P300S comparison; meanwhile, the opposite was true in the 323F3C vs. 323F3S comparison. Moreover, some DEGs in the P300C vs. P300S comparison were significantly enriched in the cell wall (GO:0005618). In terms of the molecular function, some terms related to oxidoreductase and antioxidant, such as peroxidase activity (GO:0004601), oxidoreductase activity, acting on peroxide as acceptor (GO:0016684), antioxidant activity (GO:0016209) were enriched in both 323F3 and P300 under salt stress, but the number of up-regulated genes related to these terms were higher in the P300C vs. P300S comparison than in the 323F3C vs. 323F3S comparison. We also found that many of the DEGs were significantly enriched in oxidoreductase activity (GO:0016491), transcription regulator activity (GO:0140110), and DNA–binding transcription factor activity (GO:0003700) in the P300C vs. P300S comparison.

### 2.5. Kyoto Encyclopedia of Genes and Genomes Pathway Enrichment Analysis of DEGs

Kyoto Encyclopedia of Genes and Genomes (KEGG) pathway enrichment analysis was also performed to further understand the function of these DEGs from a pathway-specific perspective (Figure 8). Our results indicated that arginine and proline metabolism (ko00330), ascorbate and aldarate metabolism (ko00053), fatty acid degradation (ko00071), glycerolipid metabolism (ko00561), isoquinoline alkaloid biosynthesis (ko00950), limonene and pinene degradation (ko00903), stilbenoid, diarylheptanoid and gingerol biosynthesis (ko00945), tropane, piperidine and pyridine alkaloid biosynthesis (ko00960), tyrosine metabolism (ko00350), and zeatin biosynthesis (ko00908) were highly enriched pathways in 323F3 under salt stress, Meanwhile, biosynthesis of unsaturated fatty acids (ko01040), circadian rhythm–plant (ko04712), cyanoamino acid metabolism (ko00460), diterpenoid biosynthesis (ko00904), monoterpenoid biosynthesis (ko00902), plant–pathogen interaction (ko04626), protein processing in endoplasmic reticulum (ko04141), sesquiterpenoid and triterpenoid biosynthesis (ko00909), starch and sucrose metabolism (ko00500), and steroid biosynthesis (ko00100) were enriched pathways in P300 under salt stress. Some pathways were commonly enriched in both 323F3 and P300 under salt stress, such as alpha-linolenic acid metabolism (ko00592), carotenoid biosynthesis (ko00906), cutin, suberine, and wax biosynthesis (ko00073), fatty acid elongation (ko00062), glutathione metabolism (ko00480), MAPK signaling pathway–plant (ko04016), nitrogen metabolism (ko00910), pentose and glucuronate interconversions (ko00040), phenylpropanoid biosynthesis (ko00940). and plant hormone signal transduction (ko04075). Moreover, we found that the number of salt–induced genes was lower than the number of salt–inhibited genes in most of these enriched KEGG pathways in 323F3, while the opposite difference was true of P300, meaning that there are more genes involved in improving salt tolerance in P300 than in 323F3.

### 2.6. Analysis of DEGs Associated with Plant Hormones

KEGG analysis indicated that some of the DEGs were enriched in the biosynthesis and signaling pathways of IAA, ETH, JA, and ABA. IAA is mainly synthesized by the TAA/YUC-dependent pathway in plants [35]. One gene (*gene–CQW23_26103*) was described as the TAA1 gene, and its expressions in 323F3 and P300 under salt stress increased 2.47–fold and 3.46–fold, respectively (Figure 9A, Appendix A), suggesting that salt stress could induce auxin synthesis in pepper roots. As for the IAA signaling pathway, there were twelve and ten relevant DEGs in the 323F3C vs. 323F3S and P300C vs. P300S comparisons, respectively (Figure 9A). Among them, four DEGs were shared in the two comparisons, including two AUX/IAA genes (*gene–CQW23_30351* and *gene–CQW23_11539*) that were up-regulated in the two comparisons, one SAUR gene (*gene–CQW23_04494*) that was down-regulated in the two comparisons, and one SAUR gene (*gene–CQW23_01382*) that was down-regulated in 323F3C vs. 323F3S and up-regulated in P300C vs. P300S. The other eight specific DEGs in 323F3C vs. 323F3S comprised one down-regulated gene that encodes GH3 protein, and two up-regulated and five down-regulated genes that encode SAUR proteins. For the other six specific DEGs in P300C vs. P300S, there was one up-regulated AUX/IAA gene (*gene–CQW23_07351*), one up-regulated ARF transcription factor (*gene–CQW23_07519*), and two up-regulated and two down-regulated SAUR genes. In summary, many IAA synthesis and signaling genes were induced by salt stress in both P300 and 323F3.

For the DEGs related to JA synthesis in P300C vs. P300S, three up-regulated DEGs, were found; these comprised one specific DEG (*gene–CQW23_11477*) that encodes LOX protein, one DEG (*gene–CQW23_20403*) that encodes OPR protein, and one specific DEG (*gene–CQW23_23376*) that encodes ACX protein, Additionally, three down-regulated DEGs were found, including one specific *LOX* gene (*gene–CQW23_01567*), one specific *AOS* gene (*gene–CQW23_23324*), and one *OPR* gene (*gene–CQW23_09013*) (Figure 9B, Appendix A). All five DEGs associated with JA synthesis, including one *AOS* gene, three *OPR* genes, and one *MFP2* gene, were down-regulated in 323F3 under salt stress. In the JA signaling pathway (Figure 9B, Appendix A), only one relevant DEG (*gene–CQW23_15121* that encodes JAZ protein) was found to be down-regulated in 323F3C vs. 323F3S, which was also down-regulated in P300C vs. P300S. Another four *JAZ* genes were found to be specific DEGs in P300C vs. P300S, and three of them (*gene–CQW23_11488*, *gene–CQW23_16942* and *gene–CQW23_06713*) were up-regulated. Moreover, one *MYC2* gene (*gene–CQW23_18667*) increased by 2.45-fold in P300 under salt stress. Briefly, all of the JA synthesis and signaling-related DEGs were down-regulated by salt in 323F3; meanwhile, in P300, several JA synthesis and signaling genes responded positively to salt stress.

In terms of the DEGs associated with ABA synthesis, one DEG (*gene–CQW23_17736*), which encodes the key synthesis protein NCED of ABA, was found to have increased 1.77-fold and 3.41-fold in the 323F3C vs. 323F3S and P300C vs. P300S comparisons, respectively (Figure 9C, Appendix A). This indicates that ABA may be involved in the response to salt stress in pepper roots. Moreover, one *β-CHY* gene (*gene–CQW23_07725*) and one *SDR* gene (*gene–CQW23_10307*) were identified as having been up-regulated under salt stress in 323F3 and P300, respectively. ABA signaling genes’ responses to salt stress were also different in the two genotypes (Figure 9C). As for the *PYR/PYL**s* that encode ABA receptors, four relevant DEGs, including three down-regulated genes and one up-regulated gene, were found in P300C vs. P300S; five relevant DEGs were found in 323F3C vs. 323F3S, all of which were down-regulated. Additionally, two *PP2C**s* (*gene–CQW23_15893* and *gene–CQW23_07281*) were clearly up-regulated under salt stress in both 323F3 and P300. In addition, we identified another three especially up-regulated DEGs (*gene–CQW23_07851*, *gene–CQW23_12866*, and *gene–CQW23_11512*) that encode PP2C proteins in 323F3C vs. 323F3S. As for the *SnRK2* and *ABF* genes, all three correlative DEGs in P300C vs. P300S were significantly down-regulated, whereas one of the two DEGs that encode SnRK2 protein and both of the two DEGs that encode ABI5 protein in 323F3C vs. 323F3S were up-regulated. In summary, ABA synthesis and signaling pathways were involved in the salt response in both 323F3 and P300.

Similarly, some of the key synthesis genes of ETH were also regulated by salt treatment in both of the two genotypes (Figure 9D, Appendix A). Two *ACS* genes (*gene–CQW23_19699* and *gene–CQW23_05330*) were especially up-regulated in P300 under salt stress. In addition, five *ACO* genes responded to salt stress in P300, including three up-regulated genes (*gene–CQW23_31171*, *gene–CQW23_01947*, and *gene–CQW23_22558*) and two down-regulated genes (*gene–CQW23_06332* and *gene–CQW23_10037*) (Figure 9D). In 323F3C vs. 323F3S, only one *ACO* gene (*gene–CQW23_01947)* was up-regulated. These results suggesting that ETH synthesis is induced more strongly by salt stress in P300 than in 323F3. For the ETH signaling genes, one up-regulated DEG that encodes CTR1 protein was found in P300C vs. P300S, whereas there was no differentially expressed *CTR1* in 323F3 under salt stress (Figure 9D). In addition, 27 DEGs, described as ERF-family genes, were identified in P300C vs. P300S, and 22 of them, including *ERF1*–*ERF5*, positively respond to salt stress in P300. Meanwhile, in 323F3C vs. 323F3S, only five of the nine DEGs that encode ERF-family proteins were up-regulated (Figure 9D). In short, salt stress activated more ethylene synthesis and signaling genes in P300 than in 323F3.

### 2.7. Analysis of DEGs Associated with ROS Scavenging

Timely ROS scavenging is very important for plant tolerance to abiotic stresses [26]. As such, we paid particular attention to the up-regulated ROS scavenging genes in 323F3 and P300 under salt stress. In total, in 323F3, one *APX* gene, two *Trx* genes, two *POD* genes, and five *GST* genes were found to be significantly up-regulated under salt stress; in P300, one *CAT* gene, one *APX* gene, one *Prx* gene, three *Trx* genes, ten *POD* genes, and fourteen *GST* genes were found to be significantly up-regulated under salt stress (Table 1). The presence of more up-regulated ROS scavenging genes in P300C vs. P300S than in 323F3C vs. 323F3S suggests that ROS scavenging pathways are more active in P300 than in 323F3 under salt stress; this is inconsistent with our suggestion that P300 is more tolerant to salinity than 323F3. Notably, the *APX2* (*gene–CQW23_21948*), *Trx-4A* (*gene–CQW23_08596*), and *GST* (*gene–CQW23_03952*) genes were highly expressed and significantly induced under salt stress in both 323F3 and P300 (Table 1); however, their transcripts were obviously higher in P300S than in 323F3S (Appendix A), suggesting that they are important for the response and tolerance to salt stress in pepper. Moreover, some genes, including the *CAT2* (*gene–CQW23_04867*), *Prx*-*2C* (*gene–CQW23_27780*), *TDX* gene (*gene–CQW23_34207*), six *POD* genes (*gene–CQW23_09144*, *gene–CQW23_08677*, *gene–CQW23_07657*, *gene–CQW23_18616*, *gene–CQW23_01286*, and *gene–CQW23_07523*), and five *GST* genes (*gene–CQW23_24335*, *gene–CQW23_25825*, *gene–CQW23_04818*, *gene–CQW23_25838*, and *gene–CQW23_17729*), were particularly up-regulated in P300C vs. P300S (Table 1), and the read counts in P300S were significantly more abundant in comparison with 323F3S (Appendix A). Thus, these DEGs might also contribute to the salt adaption of P300.

### 2.8. Analysis of Fatty Acid Desaturase and Heat-Shock-Protein-Related DEGs

Biosynthesis of the unsaturated fatty acids pathway was found to be highly enriched in P300C vs. P300S, but not enriched in 323F3C vs. 323F3S, and 12 of the 13 DEGs in this pathway responded positively to salt stress in P300 (Figure 8). Further analysis showed that 11 of the 12 up-regulated DEGs in P300C vs. P300S were annotated as Omega-6 fatty acid desaturase (FAD2) family genes, and there were no up-regulated DEGs encoding fatty acid desaturase in 323F3C vs. 323F3S (Table 2); this suggests that *FAD2* genes might be associated with their difference in salt tolerance. Notably, four *FAD2* genes (*gene–CQW23_29306*, *gene–CQW23_21403*, *gene–CQW23_29304*, and *gene–CQW23_29305*) even increased by more than twofold in P300 under salt stress (Table 2), and their transcripts in P300S were also more than twofold higher in comparison with that in 323F3S (Appendix A).

Similarly, protein processing in the endoplasmic reticulum pathway was also highly enriched in P300 under salt stress, and 17 of the up-regulated DEGs (37) involved in this pathway belonged to heat-shock protein (HSP) family genes (Figure 8 and Table 2). In total, 13 of the 17 *HSPs* were also induced by salt stress in 323F3, but most of them increased at significantly higher rates in P300 than in 323F3 under salt treatment (Table 2). Additionally, compared with 323F3S, 11 of the 17 *HSPs* showed an obviously higher expression in P300S (Appendix A). Noticeably, the *gene–CQW23_08818*, which encodes a heat-shock cognate 70 kDa protein, *gene–CQW23_14285*, which encodes a heat-shock protein 90-1, *gene–CQW23_08265*, which encodes a 17.9 kDa class II heat-shock protein, *gene–CQW23_08902*, which encodes a 15.7 kD heat-shock protein, and *gene–CQW23_00535*, which encodes a heat-shock 22 kDa protein, were highly expressed in the two genotypes under salt stress, and they were significantly up-regulated by 3.43-, 7.73-, 5.09-, 3.02-, and 2.64-fold, respectively, in 323F3C vs. 323F3S, and 8.27-, 12.91-, 10.94-, 3.19-, and 5.95-fold, respectively, in P300C vs. P300S (Table 2). Their transcripts also showed a significant and high up-regulation (more than twofold) in 323F3S vs. P300S (Appendix A), suggesting that they are related to the salt response and tolerance in pepper.

## 3. Discussion

Some pepper crops are very susceptible to salt stress, especially at the seedling growth stage, which limits their growth and productivity in many areas of the world [29]. Salt-tolerant crops are excellent gene donors for improving plant tolerance in response to salt stress; thus, it is essential to explore their salt tolerance mechanisms. In our study, we characterized the differential responses to salt stress between a salt-tolerant pepper (P300) and a salt-sensitive genotype (323F3) at the phenotypic, physiological, and transcriptomic levels. On the basis of our findings, we summarized the main salt-tolerant mechanisms in P300 from different perspectives; this should provide a basis for further research into salt resistance mechanisms and facilitate breeding for enhanced salt tolerance in pepper.

Phenotypic observations indicated that P300 showed a greater ability to resist the growth inhibition of salt stress, suggesting that P300 is more tolerant to salt stress in comparison with 323F3. The accumulation of compatible solutes is thought to be a basic strategy to protect plants from abiotic stresses [12,37]. Proline and soluble sugars are often thought of as osmoregulators in the cytoplasm of plants exposed to salt stress, which maintain appropriate cell-water status and alleviate enzyme inactivity or loss of membrane integrity due to water deficiency [14]. Goudarzi and Pakniyat (2009) reported that high proline and protein accumulations were essential to the salinity tolerance in wheat, and proline content in plants could be applied to select tolerant and susceptible genotypes under salinity stress. Our study found that P300 exhibited a lower REC than 323F3 under salt stress, meaning that P300 should more effectively maintain membrane integrity. Compared with 323F3, P300 consistently accumulated significantly higher proline and soluble sugar in its roots and leaves under salt stress. In addition, the content of soluble protein was found to be highly enhanced and accumulated in P300 roots under salt stress, which was in agreement with the high enrichment of DEGs in P300C vs. P300S in the protein processing in endoplasmic reticulum pathway. These results indicated that the high accumulation of compatible solutes under salt stress should contribute to improving the osmotic adjustment and salt tolerance of pepper—a finding supported by previous studies [30].

Maintaining ion homeostasis is highly important for plant growth under saline stress [27]. *AtNHX1*, as a dominant *cation/H+ antiporter* (*CPA*) gene involved in the SOS pathway in *Arabidopsis*, has been proven to maintain ion homeostasis under salt stress [38]. *Plasma membrane ATPase1* (*PM*-*ATPase1*) can also regulate Na^+^ accumulation in plants by providing a proton motive force that energizes the Na^+^/H^+^ antiporter [39,40]. In our study, the Na^+^ concentrations and Na^+^/K^+^ ratios in both the shoots and the roots of P300 were significantly lower than those in 323F3, suggesting that P300 should have a stronger ability to regulate ion homeostasis under salt stress. Our transcriptome data consistently indicated that the *gene–CQW23_21257* encoding CPA20 and the *gene–CQW23_00585* encoding PM-ATPase1 were specifically increased by salt stress in P300 (Figure 10A and Appendix A); meanwhile, in 323F3, although one significantly up-regulated gene (*gene–CQW23_11716*) was also described as *PM*-*ATPase* (Figure 10B), it is a low-abundance gene in pepper, regardless of salt treatment compared with the *gene–CQW23_00585* (Figure 10A and Appendix A). Taken together, these findings suggest that the SOS pathway might be more active in P300 than in 323F3 during salt stress, contributing to maintaining ion homeostasis and improving the salt tolerance in P300.

The overproduction of ROS (e.g., O_2_^−^, H_2_O_2_, and OH·) under salt stress often causes oxidative damage to enzyme activity, membrane lipids, and nucleic acids in plant cells [41]. Antioxidant enzymes have been demonstrated to be very useful in scavenging the excess ROS and protecting plant cells from oxidative damage under abiotic stresses [26,42]. SODs, such as Cu/Zn SOD, Mn SOD, and Fe SOD, can effectively scavenge superoxide anions (O_2_^−^) by converting them into H_2_O_2_ and O_2_ [43]. POD and CAT can directly scavenge H_2_O_2_ by converting it into H_2_O and O_2_ [43]. A decrease in CAT activity is thought to be the main reason for H_2_O_2_ accumulation in rice (*Oryza sativa* L.) roots [44]. Elsewhere, CAT activity was shown to increase the most drastically among the antioxidant enzymes in the roots of barley [45]. Our study indicated that the activities of the three types of antioxidant enzymes in P300 and 323F3 were consistent in their salt tolerance. In pepper leaves, their activities were induced significantly more in P300 than in 323F3. In pepper roots, their activities were also significantly increased by salt stress in P300; in 323F3, however, only POD activity was elevated, and the other two enzymes were almost unaffected under salt stress. Similarly, root RNA-seq analysis indicated that the expression of many *POD* genes was increased by salt stress in both 323F3 and P300. The *CAT2* gene was especially up-regulated by salt stress in P300, and it showed higher expression levels in the salt-exposed roots of P300 than in those of 323F3 (Appendix A). Although there was no significantly up-regulated *SOD* in the roots of P300 and 323F3 (Figure 10B and Appendix A), one *CCS* gene, which is essential for transferring copper to Cu/Zn SOD and also protects SOD from misfolding [46], was identified as especially up-regulated by salt stress in P300 roots, and was significantly up-regulated in 323F3S vs. P300S (Figure 10B and Appendix A). It has been reported that both Fe SOD and Cu/Zn SOD activity are lacking in the *CCS*-null *Arabidopsis* mutant grown in high-Cu media [47], suggesting that the *CCS* gene might be associated with the differences in the SOD activity and salt tolerance levels of 323F3 and P300.

APX is also a H_2_O_2_-scavenging enzyme that plays an important role in plant responses to environmental stresses [26]. The overexpression of *OsAPX2* in transgenic Alfalfa (*Medicago sativa* L.) showed low levels of H2O2 and enhanced salt tolerance [48]. In this study, one *APX2* was significantly up-regulated by salt stress in both 323F3 and P300, and its transcripts in P300 were about 1.4 times (<1.5 times) greater than in 323F3 under salt stress (Appendix A). Thus, we can surmise that the *APX2* is important for salt adaption in the two genotypes, although it is not the key factor that affects the differences in their salt tolerance.

GSH-dependent H_2_O_2_ metabolism is another important H_2_O_2_-scavenging process in plants; many genes, including the *GRX* and *GST* genes, are involved in the process [49]. GSTs can catalyze GSH, forming a conjugate with electrophilic compounds; GST overexpression in plants can enhanced stress tolerance [49,50,51]. GRXs function in maintaining cellular redox homeostasis and repairing oxidative damage to lipids and proteins, and play important roles in the plant abiotic stress adaptation [49]. In this study, *GRXs* were down-regulated or not affected by salt stress in 323F3; in P300, meanwhile, three up-regulated *GRXs* were identified, and two of them showed significantly higher expression in P300S compared with 323F3S (Appendix A). Similarly, more *GSTs* were up-regulated by salt stress in P300, and some of the up-regulated *GSTs* were expressed at significantly higher levels in P300 than in 323F3 under salt stress (Appendix A). All of these results suggest that GSH-dependent H_2_O_2_ metabolism should be regarded as important for the alleviation of salt-induced oxidative stress in pepper, and as contributing to the enhanced salt tolerance of P300.

The *Trx*/*Prx* system is also involved in the H_2_O_2_-scavenging process, and is important for the oxidative stress responses in plants [26,52]. The activities of SOD, POD, and CAT were enhanced in transgenic wheat by the overexpression of *TaPRX-2A* [53]. In our study, one *Prx*-*2C* gene (*gene–CQW23_27780*) was especially up-regulated by salt stress in P300, and showed a significantly higher expression in P300 than in 323F3 under salt treatment (Appendix A). Although the *Trx*-*4A* (*gene–CQW23_08596*) was up-regulated by salt stress in both P300 and 323F3, it exhibited a significantly stronger expression in P300 than in 323F3 under salt stress (Appendix A). These results suggest that high salt-induced expressions of the *Prx*-*2C* and *Trx*-*4A* genes may contribute to salt tolerance in P300.

Plant hormones, such as ETH, ABA, JA, and IAA, play important roles in plant growth, development, and environmental stress responses [19]. ETH has been considered a stress hormone, but its function in relation to salt stress is questionable [20,24]. Ethylene precursor 1-aminocyclopropane-1-carboxylic acid (ACC) can suppress the salt-sensitive phenotype in *Arabidopsis* [54]. In this study, we found that the content of ACC was significantly increased by salt stress in both 323F3 and P300; moreover, there was no significant difference in the ACC content between the two genotypes (Figure 10C), suggesting that ACC might be involved in their salt responses, but does not determine the salt tolerance in P300. Moreover, one *ACO* gene (*gene–CQW23_01947*), which converts ACC to ETH, was up-regulated in the two genotypes, supporting the idea that salt stress can induce ETH synthesis in red pepper [55]. Another *ACO* gene, named *ACO3*, was found to be especially up-regulated by salt stress in P300, and its expression inP300 was clearly higher than that in 323F3 under salt treatment (Appendix A). Recently, the mutants of *PhACO1* or *PhACO3* in petunias have been found to show significantly reduced tolerance to salt compared with WT [56], meaning that the high expression of ACO3 might contribute to improving salt tolerance in P300. *ERF* transcription factors, which are important regulatory components of ETH signaling, were highly induced by salt treatment in tomatoes [57], and the salt tolerance of tomatoes can be enhanced by the overexpression of *SlERF1* [58], *SlERF5* [59] and *SlERF84* [60]. We found that more up-regulated *ERFs* were found in P300 than in 323F3 under salt stress, suggesting that *ERFs* are involved in the salt responses of the two genotypes, and may be associated with the differences in their salt resistance. Another stress hormone is ABA. The overexpression of NCED (a key gene for ABA synthesis) in rootstocks might alleviate salinity stress in tomato shoots [61], suggesting that the root-supplied ABA functions in the salt stress response [62]. In our study, one NCED gene was found to be significantly up-regulated by salt stress in both P300 and 323F3, suggesting that ABA synthesis was induced by salt in both genotypes. *OsABI5*, an important ABA signaling gene, is increased by salt treatment, and the overexpression and repression of *OsABI5* in rice seedlings result in salt-sensitive and salt-tolerant phenotypes, respectively [63]. We found that no up-regulated *ABI5* genes existed in P300 under salt stress, whereas two *ABI5* genes were especially activated in 323F3 and their expressions in 323F3 seemed to be slightly higher than that in P300 (Appendix A), suggesting that the high sensitivity to salt stress in 323F3 might be related to *ABI5* genes that are highly expressed under salt stress. JA is also important for salt responses in plants [19]. Although we found that many genes involved in JA synthesis and signaling pathways responded positively to salt stress in P300, but not in 323F3, the transcripts of their two genotypes were generally similar under salt stress; this suggests that JA does not determine the difference in their salt tolerance levels. Auxin can regulate plant root growth and is important in helping plants to cope with salt and other abiotic stresses [19]. Our results demonstrated that many IAA synthesis and signaling genes were induced by salt stress in both 323F3 and P300, and most of them exhibited similar expression in the two genotypes under salt stress (Appendix A); this suggests that auxin is involved in the salt-stress response, but might be not related to the difference in the two genotypes’ salt tolerance levels.

The unsaturation of fatty acids in membrane lipids is also associated with the tolerance of salt stress in plants [18]. FAD2 and FAD6, two types of Omega-6 fatty acid desaturases involved in unsaturated fatty acid synthesis in the endoplasmic reticulum and plastids, respectively, have been proven to be essential for maintaining a low cytosolic Na+ concentration and improving salt tolerance in *Arabidopsis* mutants [64,65]. The *FAD6* mutant also exhibits an increased accumulation of MDA content and decreased activities of antioxidative enzymes [65]. In this study, many *FAD2* genes were identified as being especially up-regulated by salt stress in P300, and most of them were also more highly expressed in P300 than in 323F3 under salt stress (Appendix A); this finding is in agreement with the fact that P300 showed lower MDA levels, lower Na^+^ contents, and higher antioxidative enzyme activities than 323F3 under salt stress, suggesting that these factors might be involved in the improved salt tolerance of P300. HSPs, including sHsps with a molecular mass of 15 to 42 kDa, are thought to play important roles in protein processing and protect plants against abiotic stresses by preventing protein aggregation and misfolding [66]. In this study, more up-regulated *HSPs* were found in P300 than in 323F3 under salt stress, and most of the common up-regulated *HSPs* exhibited higher up-regulation by salt stress in P300 compared with 323F3 (Appendix A). In consideration of the higher content of compatible proteins in P300 than in 323F3 under salt stress, we speculate that these activated *HSPs* might help P300 cope with salt stress more effectively.

## 4. Materials and Methods

### 4.1. Plant Materials and Salt Stress Treatment

A salt-sensitive genotype (323F3) and a salt-tolerant genotype (P300) were used in this study. They were selected from the pepper germplasm resources provided by the Institute of Horticulture, Henan Academy of Agricultural Sciences, based on their performance under salt treatment at both germination and seedling stages [67,68].

The seeds sprouted at 26 °C in a growth chamber before being sown in a 72-hole tray in a greenhouse in April 2020. To eliminate the effects of salt in the soil or peat and make it easy to observe the roots, we used pure vermiculite in the tray and watered them with half-strength Hoagland’s nutrient solution (pH = 5.8) during seedling growth. After the third true leaf was completely grown, seedlings were transferred into pots (10 cm × 10 cm) only containing pure vermiculite. Seedlings were independently subjected to one of two different treatments for each genotype: no-NaCl control (323F3C and P300C, supplemented with 100 mL 1/2 strength Hoagland’s solution each time) and salt stress (323F3S and P300S, supplemented with 100 mL half-strength Hoagland’s solution and 150 mM NaCl each time), with three biological repetitions per treatment (30 plants in each replication). Salt stress treatments were performed every 3 days.

### 4.2. Physiological Analysis

After 20 days of treatment, seedlings were carefully removed and their roots were rinsed gently in water to remove residual vermiculite. Then, their phenotypes were analyzed, including plant height (PHE), stem diameter (SDA), shoot fresh and dry weight (SFW and SDW), total root length (RLE), root surface area (RSA), root diameter (RDA), root volume (RVO), root vigor (RVI), and root fresh and dry weight (RFW and RDW). PHE and SDA were measured with a ruler and digital vernier caliper, respectively. SFW, SDW, RFW, and RDW were weighed using a micrometer electronic balance. SDW and RDW were obtained by oven-drying the samples at 105 °C for 15 min, followed by 80 °C for 48 h. RLE, RSA, RDA, and RVO were measured with a root scanner (Epson, Long Beach, USA). RVI was assessed by triphenyl tetrazolium chloride (TTC) according to a previous study [69]. To evaluate the salt influences on plant growth in detail, the salt stress tolerance index of each parameter was calculated as:(Average values of NaCl-treated seedlings/Average values of control seedlings) × 100%.

To evaluate the contents of Na^+^ and K^+^, the shoots and roots of seedlings were collected separately and thoroughly dried at 80 °C for 3 days. Subsequently, 400 mg samples were ground with three replicates, and digested with 5 mL nitric acid for overnight. Then, they were kept at 80 °C for 2 h, followed by 120 °C for 2h and 160 °C for 4 h. The concentrations of Na^+^ and K^+^ were determined with an inductively coupled plasma–atomic emission spectroscopy (Thermo Fisher Scientific iCAP 7200 HS Duo, Waltham, MA, USA).

In order to assess the activities of SOD, POD, and CAT in the fresh roots and leaves (the third completely expanded leaves from plant shoot apex), about 200 mg of fresh sample was weighed with three replicates. These enzymatic antioxidant compounds (SOD, POD, and CAT) were extracted and tested with commercial kits (Suzhou Michy Biomedical Technology Co., Ltd., Suzhou, China) at absorbances of 560 nm, 470 nm, and 240 nm, respectively, according to the manufacturer’s instructions.

The free proline content was measured using acid ninhydrin colorimetry using pure proline as a standard [70]. The soluble sugar was determined by anthrone colorimetry using glucose as a standard [71]. The soluble protein content was determined using the method of Coomassie brilliant blue G-250 staining at an absorbance of 595 nm with bovine serum albumin (BSA) as a standard [72]. For their measurements, about 500 mg of fresh leaves (the third completely expanded leaves from plant shoot apex) or roots were used with three replicates. The absorbance was recorded at wavelengths of 520 nm, 625 nm, and 595 nm for proline, soluble sugar, and soluble protein, respectively, on a UV-1800 spectrophotometer.

MDA level was evaluated by the thiobarbituric acid (TBA) method [73]. About 200 mg of fresh leaves (the third totally expanded leaves from plant shoot apex) was homogenized in 5 mL of 10% trichloroacetic acid (TCA) solution for further experiments with three replicates. The absorbance of the last supernatant was recorded at wavelengths of 450, 532, and 600 nm, respectively.

To measure the electrolyte leakage, about 200 mg of fresh leaves (the third completely expanded leaves from plant shoot apex) was cut into 0.3 cm slices with three replicates and mixed with 6 mL of deionized water in a 10 mL tube. The mixture was kept at room temperature for 4 h. The electrical conductivity was first measured using a DDSSJ-3083A conductivity detector. The samples were then boiled for 20 min in a water bath. The complete conductivity was determined again in the same manner after cooling to room temperature. The relative electrical conductivity was calculated according to Song et al. (2011).

For the precise determination of endogenous ACC contents in roots, 0.1 g dry sample was weighed, ground in 1 mL water, followed by centrifugation at 14,000 g. Then, 200 μL of supernatant was taken out and mixed with 20 μL N-leucine internal standard solution in a 2 mL EP tube. Meanwhile, 200 μL ACC standard solution was used as a control. In turn, 100 μL triethylamine acetonitrile solution (PH > 7) and 100 μL phenyl isothiocyanate acetonitrile solution were added to each EP tube, followed by thorough mixing and standing for 1 h at 25 °C. Subsequently, we added 400 μL N-hexane to each EP tube, and placed them at 25 °C for 10 min. Then, 2 μL of the underlying solution was taken and diluted fivefold with water, followed by filtration with a 0.22 μm needle filter. The filtrate was carefully collected for computer tests using an Agilent1100 high-performance liquid chromatograph (HPLC) with a 254 nm wavelength.

### 4.3. RNA Extraction and Transcriptome Analysis

The root tissues of seedlings were harvested at 24 h after salt treatment and under normal growth conditions with three independent biological replicates per treatment. TRIzol reagent (Invitrogen, Carlsbad, CA, USA) was used to extract RNA according to the manufacturer’s instructions. RNA samples were reverse-transcribed using the MonScript™ RTIII All-in-One Mix with dsDNase (Monad, Wuhan, China). RNA concentrations and integrities were evaluated with a NanoDrop 2000 (Thermo Fisher Scientific, Waltham, MA, USA) and an RNA Nano 6000 Assay Kit of the Agilent Bioanalyzer 2100 system (Agilent Technologies, Santa Clara, CA, USA), respectively. RNA sequencing and assembly were carried out by Personalbio Technology Corporation (Shanghai, China). Analysis of DEGs between two samples was performed using fold change >1.5 and a *p*-value of <0.05 based on the sequencing of three independent biological replicates. GO and KEGG enrichment analyses were performed using the GOseq R package [74] and KEGG web service (https://www.kegg.jp, accessed on 15 June 2022), respectively. The threshold of *p*-value <0.05 was adopted to evaluate the significant differences in GO terms and KEGG pathways.

### 4.4. Quantitative Real-Time PCR Validation

qRT-PCR was used for the validation of the RNA-seq data using 11 different genes. Specific primers for each gene are listed in Appendix A. qRT-PCR was performed with an Applied Biosystems 7500 thermocycler (Thermo Fisher Scientific, Waltham, MA, USA) using Super-Real PreMix Plus (SYBR Green) (TIANGEN Biotech Co., Beijing, China) following the manufacturer’s instructions. The cycle conditions for qRT-PCR were: 95 °C for 3 min, followed by 40 cycles of 95 °C for 15 s and 60 °C for 40 s. *GAPDH* was used as an internal control, and the relative transcript levels were analyzed based on the 2^−(ΔΔCt)^ method [75].

## 5. Conclusions

In summary, this study discusses the different physiological and molecular mechanisms of the two contrasting pepper genotypes (323F3 and P300) in response to salt stress (Figure 11). Compared with 323F3, P300 exhibits a greater ability to resist the growth inhibition caused by salt stress. Salt stress results in cell toxicity in roots and shoots, mainly due to the over-production of ROS, the over-accumulation of Na^+^, and damage to proteins and membrane lipids. The different ROS-scavenging abilities of 323F3 and P300 may be closely related to their tolerance to salt stress. The high accumulation of compatible solutes contributes to the maintenance of cell osmotic pressure and the alleviation of enzyme inactivity. The activated expression of *HSPs* and *FAD2s* may play important roles in preventing protein and cell membrane damage. *FAD2s* may also affect Na^+^ accumulation in the cytoplasm. The increased expression of *cation/H(+) antiporter* and *plasma membrane ATPase 1* in the SOS pathway may lead to low cytosolic Na^+^ contents in P300 under salt stress. In addition, the synthesis and signaling pathways of several phytohormones were altered by salt stress, which may be involved in salt tolerance. This study shows that there are both similarities and differences in the responses of 323F3 and P300 to salt stress, and the results may provide valuable insights into the salt-response mechanisms of pepper and the hub genes for breeding strategies enhancing salt tolerance in pepper.

## Figures and Tables

**Figure 1 ijms-23-09701-f001:**
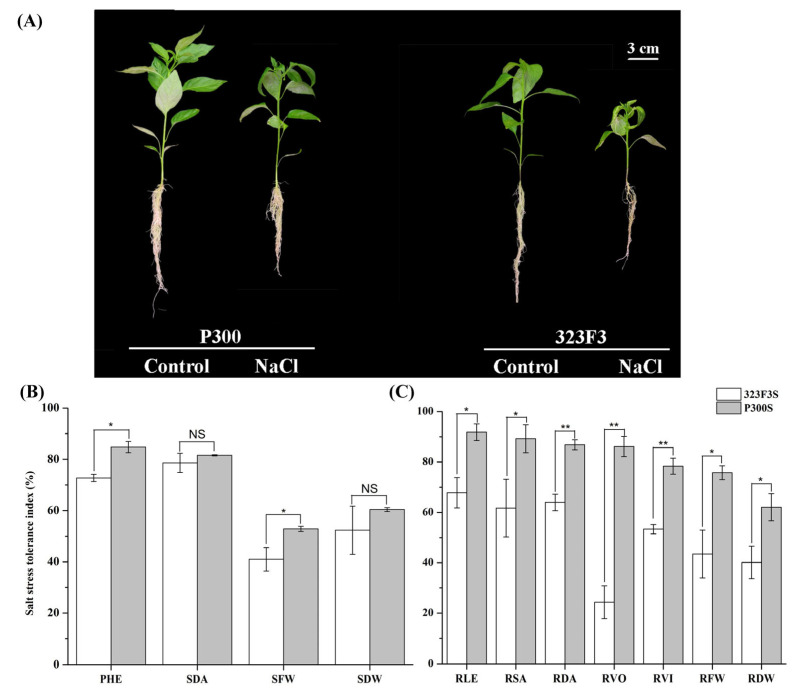
Performance of 323F3 and P300 seedlings in response to salt stress. (**A**) Phenotypes after 20 d salinity stress with 150 mM NaCl. (**B**,**C**) Salt stress tolerance index of different parameters from the shoot and root, respectively. Each bar represents three biological replicates ± SD. * *p* < 0.05; ** *p* < 0.01; NS, not significant (according to one-way ANOVA).

**Figure 2 ijms-23-09701-f002:**
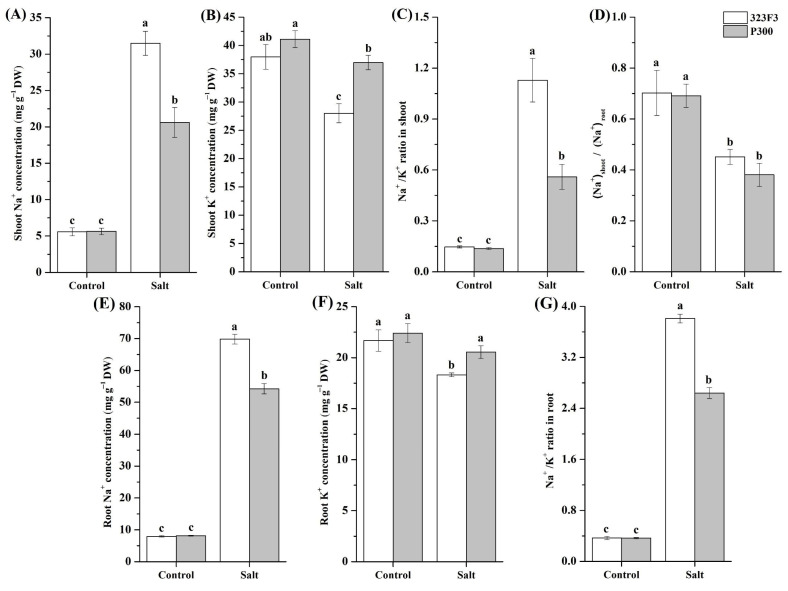
Shoot and root Na^+^ and K^+^ concentrations and Na^+^/K^+^ ratio in 323F3 and P300 under control and salt treatment conditions. Each bar represents three biological replicates ± SD. Different small letters represent significant differences at *p* < 0.05 according to one-way ANOVA. (**A**–**C**) Na^+^ and K^+^ concentrations and Na^+^/K^+^ ratio in the shoots of 323F3 and P300. (**D**) The ratio of Na^+^ concentration in shoots and roots. (**E**–**G**) Na^+^ and K^+^ concentrations and Na^+^/K^+^ ratio in the roots of 323F3 and P300.

**Figure 3 ijms-23-09701-f003:**
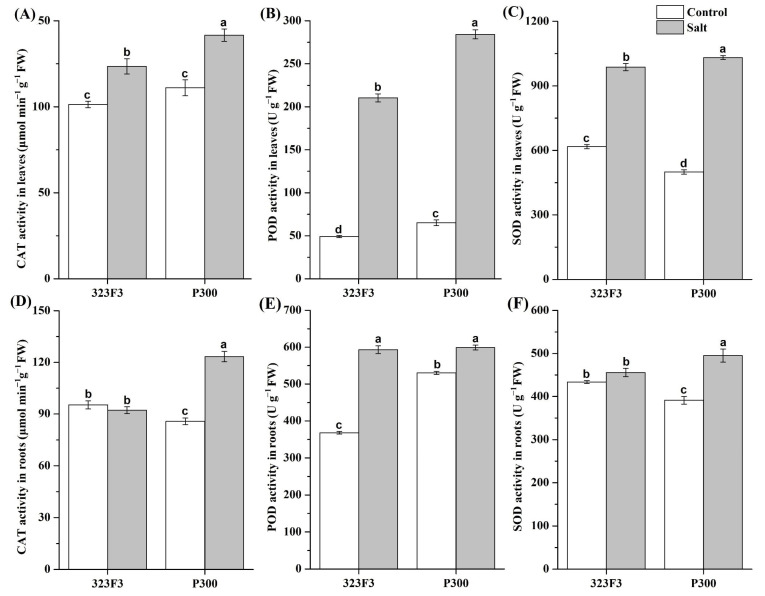
Effect of salt treatment on the antioxidase activities in the leaves (**A**–**C**) and roots (**D**–**F**) of 323F3 and P300. Each bar represents three biological replicates ± SD. Different small letters represent significant differences at *p* < 0.05 according to one-way ANOVA.

**Figure 4 ijms-23-09701-f004:**
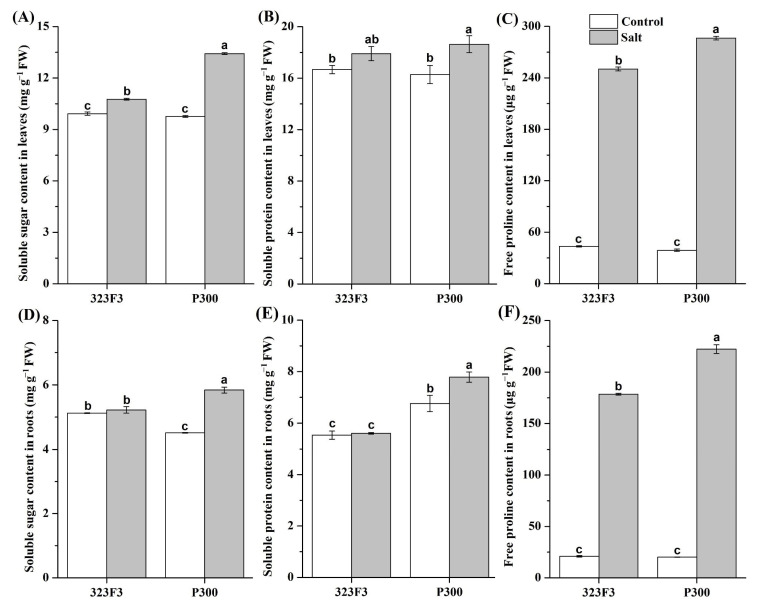
Accumulation of soluble sugar, soluble protein, and proline in the leaves (**A**–**C**) and roots (**D**–**F**) of 323F3 and P300. Each bar represents three biological replicates ± SD. Different small letters represent significant differences at *p* < 0.05 according to one-way ANOVA.

**Figure 5 ijms-23-09701-f005:**
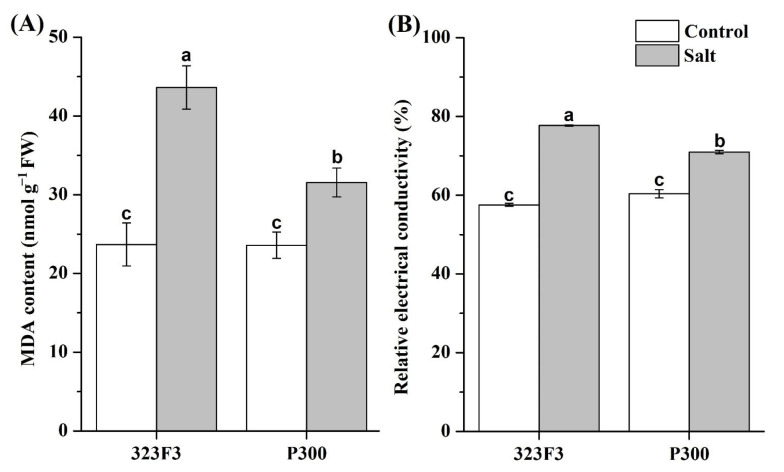
MDA content (**A**) and relative electrical conductivity (**B**) in the leaves of 323F3 and P300. Each bar represents three biological replicates ± SD. Different small letters represent significant differences at *p* < 0.05 according to one-way ANOVA.

**Figure 6 ijms-23-09701-f006:**
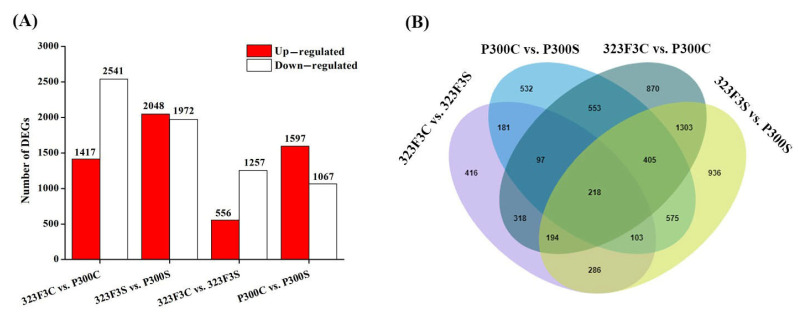
DEGs identified in the four comparisons among the control and salt treatment groups. (**A**) Total number of the up-regulated and down-regulated DEGs in the four comparisons. (**B**) Venn diagram of all DEGs in the four comparisons.

**Figure 7 ijms-23-09701-f007:**
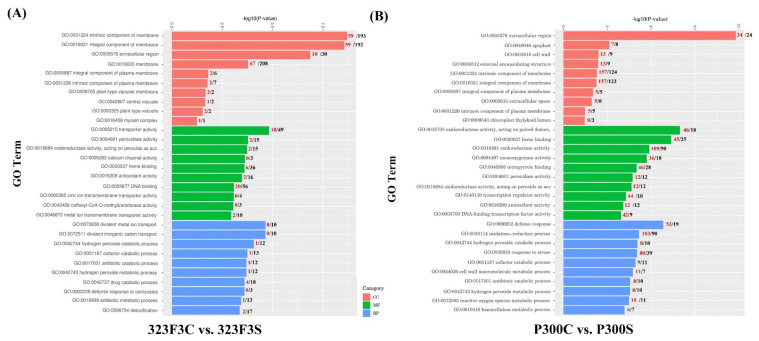
GO enrichment analysis of the DEGs identified in the 323F3C vs. 323F3S (**A**) and P300c vs. P300S (**B**) comparisons. The topmost enriched GO terms under the three main GO categories are shown. The numbers of the up-regulated and down-regulated genes in each category are represented by red and black numbers, respectively. CC, cellular component; MF, molecular function; BP, biological process.

**Figure 8 ijms-23-09701-f008:**
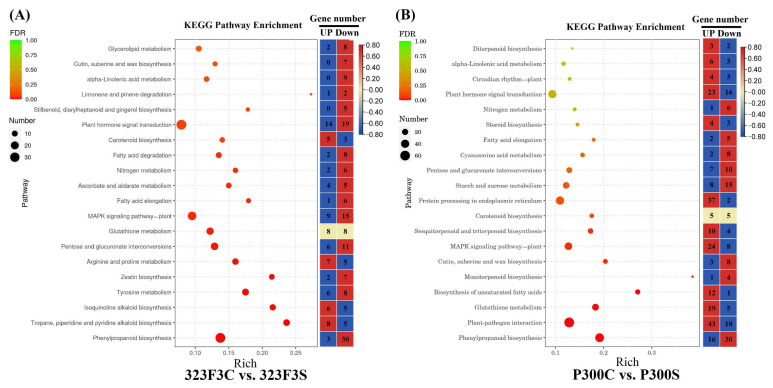
KEGG pathway enrichment analysis of the DEGs identified in the 323F3C vs. 323F3S (**A**) and P300C vs. P300S (**B**) comparisons. The topmost enriched 20 KEGG pathways with the lowest FDR values were selected for display. Rich factor refers to the ratio of the DEGS to the total number of annotated genes in each pathway. The numbers of up-regulated and down-regulated genes in each pathway are shown on the right side of each KEGG enrichment diagram.

**Figure 9 ijms-23-09701-f009:**
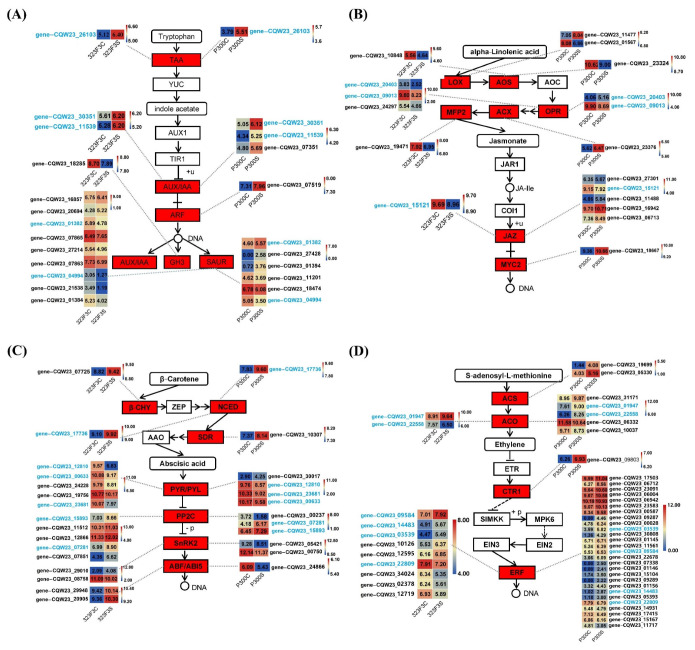
DEGs in the 323F3C vs. 323F3S and P300C vs. P300S comparisons annotated in the hormone synthesis and signaling transduction pathways by KEGG analysis. The main synthesis and signal transduction pathway of Auxin/IAA, JA, ABA, and ETH are shown in (**A**–**D**), respectively. Log2 (read counts + 1) of each gene in different samples are shown in the appropriate grid. Heat-maps were drawn using Tbtools software [36].

**Figure 10 ijms-23-09701-f010:**
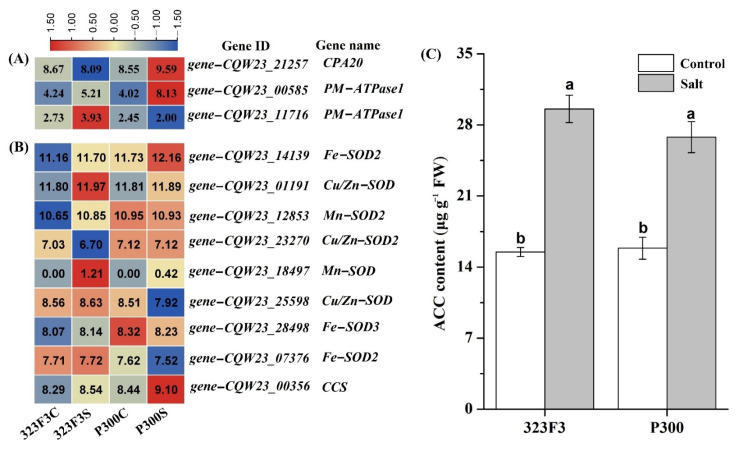
Heatmap analysis of the gene expressions and ACC content comparisons in 323F3 and P300. (**A**) The genes encoding SOD-type proteins and copper chaperone for SOD (CCS). (**B**) The DEGs involved in SOS pathway in the 323F3C vs. 323F3S and P300c vs. P300S comparisons. Log2 (read counts + 1) of each gene in different samples are shown in the heatmaps and the values of each gene in different samples were scaled by normalization using Tbtools software. (**C**) ACC content determination in 323F3 and P300 under control and salt conditions. Different small letters represent significant differences at *p* < 0.05 according to one-way ANOVA.

**Figure 11 ijms-23-09701-f011:**
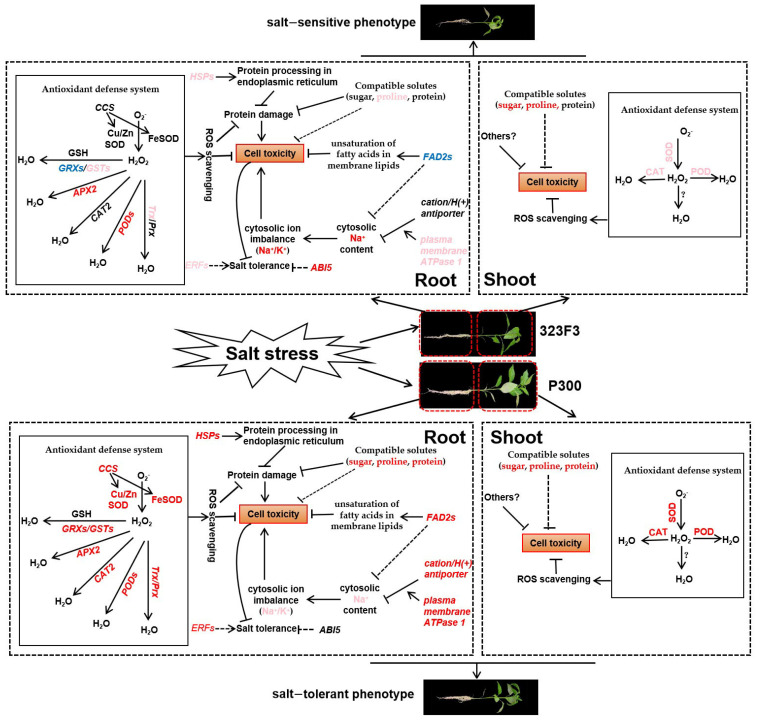
Proposed model illustrating the main salt tolerance mechanisms in seedlings of the two contrasting pepper genotypes (the salt-tolerant genotype P300 and the salt-sensitive genotype 323F3). Red fonts indicate strong up-regulation, pink fonts indicate weak up-regulation, green fonts indicate down-regulation when exposed to salinity in the two genotypes compared with their controls.

**Table 1 ijms-23-09701-t001:** The up-regulated genes associated with ROS scavenging in 323F3C vs. 323F3S and P300C vs. P300S.

Protein Type	Gene ID	Gene Description	Gene Name	323F3C vs. 323F3S		P300C vs. P300S	
				Fold	*p*-Value	Fold	*p*-Value
CAT	*gene–CQW23_04867*	Catalase isozyme 2	*CAT2*	1.24	3.99 × 10^−3^	1.88↑	8.86 × 10^−25^
APX	*gene–CQW23_21948*	L-ascorbate peroxidase 2, cytosolic	*APX2*	2.00↑	1.95 × 10^−29^	1.54↑	5.71 × 10^−28^
Prx	*gene–CQW23_27780*	Peroxiredoxin-2C	*Prx-2C*	1.02	0.8449117	21.79↑	4.86 × 10^−208^
Trx	*gene–CQW23_08596*	Thioredoxin-like protein 4A	*Trx-4A*	2.03↑	3.18 × 10^−8^	2.07↑	1.22 × 10^−18^
	*gene–CQW23_23211*	TPR repeat-containing thioredoxin TTL1	*TTL1*	1.62↑	3.72 × 10^−3^	1.01	0.96941233
	*gene–CQW23_34207*	TPR repeat-containing thioredoxin TDX	*TDX*	0.78	0.6865073	11.13↑	1.86 × 10^−4^
	*gene–CQW23_18479*	TPR repeat-containing thioredoxin TDX	*TDX*	0.72	0.2916553	7.81↑	8.08 × 10^−3^
POD	*gene–CQW23_09144*	Suberization-associated anionic peroxidase 1	*POD1*	0.59↓	6.22 × 10^−10^	2.34↑	1.65 × 10^−60^
	*gene–CQW23_06899*	Peroxidase 28	*POD28*	1.51↑	8.34 × 10^−9^	0.71	1.54 × 10^−5^
	*gene–CQW23_32462*	Peroxidase 4	*POD4*	1.79↑	4.96 × 10^−10^	6.69↑	5.17 × 10^−14^
	*gene–CQW23_05116*	Peroxidase 15	*POD15*	1.01	0.8866078	2.04↑	1.0371 × 10^−52^
	*gene–CQW23_08677*	Peroxidase 5	*POD5*	1.41	1.44 × 10^−5^	2.16↑	4.59 × 10^−44^
	*gene–CQW23_07657*	Peroxidase 4	*POD4*	1.19	0.7675621	3.34↑	0.02781563
	*gene–CQW23_18616*	Peroxidase 6	*POD6*	1.10	0.858594	1.95↑	0.0165733
	*gene–CQW23_01286*	Lignin-forming anionic peroxidase	*POD*	0.81	0.0972802	2.44↑	1.20 × 10^−3^
	*gene–CQW23_07523*	Peroxidase 66	*POD66*	1.18	0.1376819	2.10↑	5.91 × 10^−12^
	*gene–CQW23_10318*	Peroxidase 12	*POD12*	1.12	2.42 × 10^−3^	1.63↑	2.77 × 10^−28^
	*gene–CQW23_16671*	Peroxidase 72	*POD72*	1.13	0.5396232	Inf ^a^↑	3.52 × 10^−23^
GRX	*gene–CQW23_02281*	Glutaredoxin-C1	*GRX-C1*	0.45↓	6.30 × 10^−3^	4.22↑	3.23 × 10^−7^
	*gene–CQW23_22057*	Monothiol glutaredoxin-S2	*GRX-S2*	0.59	0.3190946	2.78↑	1.93 × 10^−3^
	*gene–CQW23_17543*	Glutaredoxin-C9	*GRX-C9*	0.68	3.47 × 10^−3^	1.62↑	0.01676165
GST	*gene–CQW23_03952*	Putative glutathione S-transferase	*GST*	1.57↑	3.96× 10^−22^	3.98↑	2.55 × 10^−224^
	*gene–CQW23_21928*	Putative glutathione S-transferase	*GST*	1.68↑	2.34 × 10^−6^	3.04↑	5.79 × 10^−16^
	*gene–CQW23_21929*	Putative glutathione S-transferase	*GST*	1.64↑	0.0198286	2.63↑	1.56 × 10^−6^
	*gene–CQW23_22418*	Glutathione S-transferase U9	*GST-U9*	1.93↑	6.24 × 10^−30^	4.57↑	5.97 × 10^−145^
	*gene–CQW23_25828*	Glutathione S-transferase U18	*GST-U18*	1.71↑	1.46 × 10^−74^	1.09	1.99 × 10^−3^
	*gene–CQW23_24335*	Putative glutathione S-transferase	*GST*	0.81	3.94 × 10^−4^	2.54↑	5.87 × 10^−45^
	*gene–CQW23_25825*	Glutathione S-transferase U18	*GST-U18*	0.78	3.26 × 10^−4^	2.01↑	6.59 × 10^−23^
	*gene–CQW23_21930*	Putative glutathione S-transferase	*GST*	1.45	1.55 × 10^−13^	1.86↑	2.62 × 10^−15^
	*gene–CQW23_04818*	Putative glutathione S-transferase	*GST*	0.28	0.6107545	19.02↑	1.38 × 10^−10^
	*gene–CQW23_25838*	Glutathione S-transferase U18	*GST-U18*	0.39	0.4708276	12.18↑	2.22 × 10^−9^
	*gene–CQW23_22421*	Glutathione S-transferase U9	*GST-U9*	1.28	4.42 × 10^−3^	1.81↑	5.99 × 10^−9^
	*gene–CQW23_17729*	Putative glutathione S-transferase	*GST*	0.48	1	13.81↑	1.63 × 10^−7^
	*gene–CQW23_02556*	Putative glutathione S-transferase	*GST*	1.23	0.1406508	2.01↑	3.56 × 10^−7^
	*gene–CQW23_21533*	Putative glutathione S-transferase	*GST*	1.61	0.0656395	3.64↑	5.19 × 10^−5^
	*gene–CQW23_21187*	Putative glutathione S-transferase	*GST*	1.03	0.9176443	1.99↑	0.03055018

Note: expression patterns: “↑” is up-regulated; “↓” is down-regulated. The read counts of each gene in different samples can be found in Appendix A. ^a^ A gene could not be detected in the P300C sample, but highly expressed in the P300S sample.

**Table 2 ijms-23-09701-t002:** The up-regulated genes encoding HSP and FAD type proteins in 323F3C vs. 323F3S and P300C vs. P300S.

Protein Type	Gene ID	Gene Description	Gene Name	323F3C vs. 323F3S		P300C vs. P300S	
				Fold	*p*-Value	Fold	*p*-Value
HSP	*gene–CQW23_08818*	Heat-shock cognate 70 kDa protein	*HSP70*	3.43↑	0	8.27↑	0
	*gene–CQW23_14285*	Heat-shock protein 90-1	*HSP90-1*	7.73↑	0	12.91↑	0
	*gene–CQW23_15919*	Heat-shock cognate 70 kDa protein	*HSP70*	2.37↑	0	2.74↑	0
	*gene–CQW23_08264*	17.9 kDa class II heat-shock protein	*HSP17.9*	3.79↑	1.09 × 10^−59^	7.09↑	1.91 × 10^−112^
	*gene–CQW23_00534*	Heat-shock 22 kDa protein, mitochondrial	*HSP22*	2.60↑	1.04 × 10^−50^	6.99↑	6.98 × 10^−126^
	*gene–CQW23_08265*	17.9 kDa class II heat-shock protein	*HSP17.9*	5.09↑	2.16 × 10^−21^	10.94↑	6.73 × 10^−123^
	*gene–CQW23_08902*	15.7 kDa heat-shock protein, peroxisomal	*HSP15.7*	3.02↑	4.62 × 10^−21^	3.19↑	3.39 × 10^−45^
	*gene–CQW23_31624*	18.5 kDa class I heat-hock protein	*HSP18.5*	2.02↑	1.17 × 10^−19^	4.86↑	2.08 × 10^−100^
	*gene–CQW23_11457*	17.4 kDa class III heat-shock protein	*HSP17.4*	4.90↑	9.92 × 10^−17^	5.01↑	6.07 × 10^−17^
	*gene–CQW23_00535*	Heat-shock 22 kDa protein, mitochondrial	*HSP22*	2.64↑	7.74 × 10^−14^	5.95↑	1.085 × 10^−87^
	*gene–CQW23_08718*	22.0 kDa class IV heat-shock protein	*HSP22*	2.01↑	5.70 × 10^−4^	1.69↑	2.88 × 10^−5^
	*gene–CQW23_31623*	18.5 kDa class I heat-shock protein	*HSP18.5*	2.68↑	1.03 × 10^−3^	2.31↑	2.09 × 10^−4^
	*gene–CQW23_11673*	Heat-shock 70 kDa protein	*HSP70*	1.99↑	0.015001	2.44↑	0.02448657
	*gene–CQW23_23188*	Heat-shock protein 90-2	*HSP90-1*	1.37	4.04 × 10^−76^	1.58↑	6.33 × 10^−106^
	*gene–CQW23_24043*	Heat-shock cognate 70 kDa protein	*HSP70*	1.29	4.27 × 10^−45^	1.51↑	6.56 × 10^−84^
	*gene–CQW23_31620*	18.5 kDa class I heat-shock protein	*HSP18.5*	1.02	0.9619644	2.14↑	1.43 × 10^−3^
	*gene–CQW23_01345*	26.5 kDa heat-shock protein, mitochondrial	*HSP26.5*	1.54	0.4367286	3.02↑	0.01976457
FAD	*gene–CQW23_29307*	Omega-6 fatty acid desaturase, endoplasmic reticulum	*FAD2*	0.16↓	0	1.62↑	0.02217258
	*gene–CQW23_29533*	Omega-6 fatty acid desaturase, endoplasmic reticulum	*FAD2*	0.60↓	3.46 × 10^−8^	1.62↑	7.31 × 10^−5^
	*gene–CQW23_29306*	Omega-6 fatty acid desaturase, endoplasmic reticulum	*FAD2*	0.18↓	6.23 × 10^−3^	29.17↑	1.12 × 10^−10^
	*gene–CQW23_29537*	Omega-6 fatty acid desaturase, endoplasmic reticulum	*FAD2*	0.77	7.26 × 10^−31^	2.06↑	5.75 × 10^−136^
	*gene–CQW23_21403*	Omega-6 fatty acid desaturase, endoplasmic reticulum	*FAD2*	1.08	0.3271504	3.77↑	9.26 × 10^−67^
	*gene–CQW23_29304*	Omega-6 fatty acid desaturase, endoplasmic reticulum	*FAD2*	0.84	0.2026156	4.04↑	1.28 × 10^−36^
	*gene–CQW23_35257*	Omega-6 fatty acid desaturase, endoplasmic reticulum	*FAD2*	0.79	3.40 × 10^−14^	1.55↑	1.01 × 10^−20^
	*gene–CQW23_29539*	Omega-6 fatty acid desaturase, endoplasmic reticulum	*FAD2*	0.74	1.30 × 10^−6^	1.73↑	6.26 × 10^−15^
	*gene–CQW23_29540*	Omega-6 fatty acid desaturase, endoplasmic reticulum	*FAD2*	0.89	0.0626088	1.75↑	6.35 × 10^−9^
	*gene–CQW23_29534*	Omega-6 fatty acid desaturase, endoplasmic reticulum	*FAD2*	0.94	0.6231729	1.69↑	3.66 × 10^−8^
	*gene–CQW23_29305*	Omega-6 fatty acid desaturase, endoplasmic reticulum	*FAD2*	0.70	0.6677045	11.45↑	2.39 × 10^−5^

Note: expression patterns: “↑” is up-regulated; “↓” is down-regulated. The read counts of each gene in different samples can be found in Appendix A.

## Data Availability

The raw data for this study can be found in the National Center for Biotechnology Information (NCBI) repository, bioproject: PRJNA837070.

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
