# Peer review of "Comparative Physiological and Transcriptomic Analyses of Two Contrasting Pepper Genotypes under Salt Stress Reveal Complex Salt Tolerance Mechanisms in Seedlings"

_ijms, 2022, doi:10.3390/ijms23179701_

Round 1

Reviewer 1 Report

The authors propose a manuscript titled “Comparative Physiological and Transcriptomic Analyses of Two Contrasting Pepper Genotypes Under Salt Stress Reveal  Complex Salt Tolerance Mechanisms in Seedling”. The authors discuss on on a crucial topic related on pepper (Capsicum annuum L.) that is widely cultivated worldwide, evaluating the salinity damage, especially at the seedling stage. In particular the researchs was conducted on physiological and transcriptional differences between two genotype seedlings (P300 and 323F3).  Many differences was discovered between these two genotypes. In the work is also found differences in the hormone synthesis and signaling pathway genes in both P300 and 323F3 under salt stress. The work suggest some candidate genes for improving salt tolerance in pepper.

The manuscript is original in the data compared to other similar articles. However I believe it is necessary to implement the maniscript with some crucial concepts that the authors will have no problem to accepting as they are designed to improve the work.

1. Introduction

Some periods deserve to literature data and please give a number not directly the reference. See guidelines of Journal.

·      Lines 36-37. With the constantly increasing world population, improving plant salt tolerance to meet the growing demand for food is essential [choose reference];

·      Lines 37-40. Please complete the period with a crucial concept from CWR, in this way: “Investigating the mechanisms of crop responses to salt stress could improve understanding of the genetic basis of salt tolerance and provide the basis for effective engineering strategies to improve salt tolerance [2], starting from the reconsideration of wild donors, called Crop Wild Relatives (CWRs) [e.g. Perrino and Wagensommer 2022, Abenavoli et al. 2021 and choose other 2 references on this crucial topic];

·      Lines 43-45. To prevent growth cessation or cell death, plants adopt many adaptive mechanisms to confront salt stess [choose references].

·      Lines 53-54. Remember for botanical point of view the scientific plant name must be reported in the complete way, with the authors that discovered for the first time the species. Check whole document and correct accordingly. I suggest this site: http://ww2.bgbm.org/IOPI/gpc/default.asp

ü Arabidopsis thaliana Heynh.

·      Line 57. See my previous comment.

ü Glycine max (L.) Merr.;

·      Line 82. Capsicum annuum L. in italic

Reference to be added:

ü Abenavoli, L.; Milanovic, M.; Procopio, A.C.; Spampinato, G. et al. Ancient wheats: beneficial effects on insulin resistance. Minerva Medica 2021112, 641-50. doi: 10.23736/S0026-4806.20.06873-1

ü Perrino, E.V.; Wagensommer, R.P. Crop Wild Relatives (CWRs) Threatened and Endemic to Italy: Urgent Actions for Protection and Use. Biology 2022, 11, 193. https://doi.org/10.3390/biology11020193

2. Results

Well done. Few suggestions. The tables are clear.

·        Figure 1. I suggested to increase the size of the picture (A);

·        Figure 7,8, and 9. Please increase the size of this figure are unclear;

3. Discussion

Well done. No suggestions. 

4. Materials and Methods

Weel done, Only a consideration. 

·      Line 589-591. The authors eclar that the material selected derive from the pepper germplasm resources provided by the Institute of Horticulture, Henan Academy of Agricultural Sciences. The question is: were also evaluated were the conditions in the open or experimental field outside the institute?

Conclusion

Please spend two more words on results and prospectives.

References

Please give DOI when is available

Author Response

Response 1: We are grateful for your constructive and insightful criticism and advice about the reference. We have added relevant literature according to your suggestions, which makes the introduction more substantial, reasonable and persuasive. Moreover, we have given a number of the cultivated land proportion affected by salt stress worldwide according to previous literature, which can fully show out that salinity is one of the most brutal environmental stresses.

       We agree with the comments that the scientific plant name must be reported in the complete way. We have checked the whole document carefully and corrected accordingly in the manuscript.

Point 2: Results

Well done. Few suggestions. The tables are clear.

  • Figure 1. I suggested to increase the size of the picture (A);
  • Figure 7,8, and 9. Please increase the size of this figure are unclear;

Response 2: Thank you for your suggestions. We have increased the size of the picture (A) in Figure 1 and the size of Figure 7,8, and 9 appropriately to make them clearer.

Point 3: Materials and Methods

Weel done, Only a consideration. 

  • Line 589-591. The authors declare that the material selected derive from the pepper germplasm resources provided by the Institute of Horticulture, Henan Academy of Agricultural Sciences. The question is: were also evaluated were the conditions in the open or experimental field outside the institute?

Response 3: We are glad to answer this question. In fact, we have given a comprehensive evaluation on the salt tolerance of more than 100 pepper germplasm resources in the experimental field outside the institute before, and the results have been published in the Journal of Henan Agricultural University and the Shandong Agricultural Sciences. The two references have been added in this paper.

Point 4: Conclusion

Please spend two more words on results and prospectives.

Response 4: Thank you for your valuable and thoughtful suggestions which helped us to improve the conclusion in this paper. We have added the results about the performance of the two genotypes (323F3 and P300) under salt stress and several words on perspectives in the conclusion. For example, we have turned “the two contrasting pepper genotypes (323F3 and P300) in response to salt stress” into “the two contrasting pepper genotypes (323F3 and P300) in response to salt stress”. We have added the sentence “Compared to 323F3, P300 exhibits a greater ability to resist the growth inhibition caused by salt stress.” followed by the first sentence in the conclusion. We also have turned the last sentence “the results may provide valuable insights into the salt response mechanisms of pepper and hub genes enhanced salt tolerance in pepper” into “the results may provide valuable insights into the salt-response mechanisms of pepper and the hub genes for breeding strategies enhancing salt tolerance in pepper”

Point 5: References

Please give DOI when is available

Response 5: Thank you for your advice. As your suggestion, we have added DOI to all of the references in this manuscript, except the two references (27 and 28) which have no available DOI.  

Reviewer 2 Report

This is a solid and well-designed project.  I have only a few minor suggestions:

The caption to figure 4 mentions panels A-C refer to the leaves and B-D refer to the roots.  It appears that D-F refer to the roots.

What threshold is being used to define "up-regulated" and "down-regulated" for genes/pathways?

Line 635: please provide the make of the spectrophotometer.

Author Response

Point 1: The caption to figure 4 mentions panels A-C refer to the leaves and B-D refer to the roots.  It appears that D-F refer to the roots.

Response 1: We agree with the comments and have revised the caption to figure 4 in the manuscript.

Point 2: What threshold is being used to define "up-regulated" and "down-regulated" for genes/pathways?

Response 2: We are glad to answer this question. In this study, fold change > 1.5 and p-value <0.05 were used to selected DEGs in different comparisons, and The threshold of p-value < 0.05 was adopted to evaluate the significant differences in GO terms and KEGG pathways, which have been shown in the materials and methods. Up-regulated means more highly expressed compared to the reference, and down-regulated means expressed lower compared to the reference.

Point 3: Line 635: please provide the make of the spectrophotometer.

Response 3: Thank you for your advice. We have provided the absorbance (595 nm) used in the determination of soluble protein content on a UV-1800 spectrophotometer.
